# A New Strategy for the Use of Post-Processing Vacuum Bags from Aerospace Supplies: Nucleating Agent to LLDPE Phase in PA6/LLDPE Blends

**Gabriel Portilho Monteiro de Souza [1], Erick Gabriel Ribeiro dos Anjos [1], Larissa Stieven Montagna [1], Orestes Ferro [2] and Fabio Roberto Passador [1,\*]**

[1] Polymer and Biopolymer Technology Laboratory (TecPBio), Department of Science and Technology, Federal University of São Paulo (UNIFESP), 330 Talim St., São José dos Campos 12231-280, SP, Brazil; gsouza091@gmail.com (G.P.M.d.S.); erickgabsjc@gmail.com (E.G.R.d.A.); larissambiental@yahoo.com.br (L.S.M.)

[2] ALLTEC Materiais Compostos Ltda, 456 Moxotó St., São José dos Campos 12238-320, SP, Brazil; orestes.ferro@allteccomposites.com.br

\* Correspondence: fabiopassador@gmail.com

**Abstract:** In the aerospace industry, many composite parts are manufactured by processes using plastic vacuum bags made of polyamide 6 (PA6) as a consumable material. This implies that after demolding the part, this plastic material should be discarded, generating a considerable amount of waste. Tons of vacuum bags are discarded and incinerated per month by several companies in this sector, which highlights the need to recycle and/or reuse this material. PA6/linear low-density polyethylene (LLDPE) blends are of great technological interest because they can combine the excellent thermo-mechanical and oxygen barrier properties of the PA6 with high impact strength, good processability, and low cost of LLDPE. The replacement of neat PA6 by the post-processing vacuum bags residue PA6 may be a new strategy for the recycling of this material. In this work, PA6/LLDPE/maleic anhydride-grafted LLDPE (LLDPE-*g*-MA) (90/5/5) blends were prepared using a co-rotational twin-screw extruder and the neat PA6 was replaced by different contents of post-processing PA6 (5, 10, 15 and 20 wt.%). The mechanical, thermal, and morphological characterization was evaluated. The increase in the content of post-processing PA6 caused an increase in the crystallinity degree of the LLDPE phase, acting as a nucleating agent to the LLDPE phase, reducing the toughening effect of this phase in the blends and, therefore, providing this phase to act as a reinforcing agent.

**Keywords:** blends; PA6; LLDPE; post-processing PA6; recycling; vacuum-bag

## 1. Introduction

The increase of consumption of polymeric materials in various sectors of the industry and in the most diverse applications generates an expressive amount of post-processing waste, which can cause environmental problems making it necessary to adopt appropriate measures to reduce these problems [1,2]. The recycling of these plastics wastes is one of the most feasible ways to overcome these problems, because it allows combining environmental interests with economic benefits [3,4]. In this context, the aerospace industry uses a large quantity of thermoplastic materials in the form of vacuum bags, as consumable materials, for manufacturing aircraft composite parts. Generally, a vacuum bag is nothing more than a plastic film assembled over a composite material based on thermosetting resin with fibrous reinforcement and submitted to vacuum. This procedure, so called vacuum bagging, exists in order to mold correctly the part over the mold and remove entrapped air and gases from the part that could cause internal or superficial voids in it. The vacuum bags are composed mostly by polyamide 6

(PA6) and usually are used only in a single heating cycle inside an oven or autoclave. Subsequently to the part demolding, this plastic film is discarded and/or incinerated. For instance, Alltec Materiais Compostos Ltda., a midsize company located in São José dos Campos (Brazil), manufactures several composite parts for the aircraft industry and estimates that about 1.5 tons of this type of material are discarded and incinerated per month. Considering the other companies in this sector, this number becomes much larger, which shows, therefore, the need to recycle this material for both environmental and economic purposes.

PA6 is an engineering thermoplastic polymer that has excellent thermomechanical properties, good chemical resistance in organic media, and low oxygen permeability. However, some of its inherent characteristics such as high water absorption, fast crystallization, low dimensional stability, and very low notched impact resistance may limit its use [5,6]. One way to improve the performance of PA6 is the addition of other phases in the composition by physical mixture, for example [7]. Polyolefins like polyethylene (PE) [6,8–15] and polypropylene (PP) [16–18] are the most widely used materials to produce a polymer blend with PA6 because these materials have high impact strength, good processability, low water absorption, and low cost [6,13,16–18]. In this way, PA6/ linear low-density polyethylene (LLDPE) blends allow us to combine the properties of these polymers and expand the possibilities of applications of these materials. Shin and Han [8] prepared PA6/LLDPE/Glycidyl methacrylate (GMA) blends using an electron-beam initiated mediation process. The materials were prepared using a twin-screw co-rotating extruder. The authors observed an improvement in mechanical properties. The blend irradiated with 100 kGy presented an elongation at break of approximately four times higher than that of neat PA6. Shi et al. [13] prepared LLDPE/PA6 blends with low PA6 contents and the addition of maleic anhydride functionalized polybutadiene (PB-*g*-MA) via reactive extrusion. It was observed that the formation of the copolymer LLDPE-*g*-PB-*g*-MA, which strengthened the interface and favored stress transfer, allowed PA6 phase deformation in the blends. It was also observed the formation of the PB-*g*-PA6 copolymer that facilitated the formation of a flat interface between LLDPE and PA6. As a result, PA6 phase elongation was achieved, and it was possible to obtain the co-continuous morphology, which gives the maximum contribution of the elastic modulus of each component and yields a synergistic effect on the impact properties.

However, the mixture of PA6 with polyethylenes presents a great challenge because the polyamide has polar groups that do not interact with the nonpolar groups of polyethylene, making the mixture immiscible and incompatible [6,8–15]. Therefore, it is necessary to use a compatibilizer agent, which are generally copolymers or polymers grafted with reactive groups of different polarity from one of the phases of the blend, for example, maleic anhydride-grafted linear low-density polyethylene (LLDPE-*g*-MA). This compatibilizer agent is miscible in the LLDPE phase and its anhydride groups can react and/or interact with the amine groups of PA6 to form the imide group. In this way, the LLDPE-*g*-MA-*g*-PA6 copolymer can be obtained, increasing the interfacial adhesion between the PA6 and LLDPE phases and, thus, improving the mechanical properties of the blend [6,11]. Das et al. [6] prepared PA6/LLDPE blends compatibilized with PE-*g*-MA. The authors observed that the addition of PE-*g*-MA compatibilizer caused a decrease in the particle size of the second phase and an improvement of the mechanical properties and water absorption behavior. Yoon et al. [19] used high frequency welding to prepare LLDPE/PA6 blends compatibilized with maleic anhydride-grafted polyethylene (PE-*g*-MA). It was observed that the use of high frequency welding was only possible due to the use of the compatibilizer agent. Kelar e Jurkowski [20] studied LDPE/PA6 blends compatibilized with LDPE-*g*-MA and verify the morphology and mechanical properties. The authors observed an increase in the compatibility of the polymer phases and in the ultimate tensile strength of the compatibilized blends. These studies show that PA6/LLDPE blends are of great interest and are therefore the subject of this study.

Therefore, in this work the post-processing PA6 was used instead of neat PA6 for the preparation of PA6/LLDPE blends. The new strategy of mixture consists of the cryogenic grinding of post-processing PA6 from vacuum bags, subsequent drying, and addition of this material directly to the extruder to

prepare the polymer blends. Another goal is to study the effect of the addition of different post-processing PA6 contents (5, 10, 15, and 20 wt.%) on the thermal, mechanical, and morphological properties of the PA6/LLDPE blends. In addition, it is expected that this work will determine the content of post-processing PA6 that can be used without significant losses of PA6/LLDPE blends properties.

## 2. Experimental Section

### 2.1. Materials

The polyamide 6 (PA6) of specification LIBOLON N400, with a density of 1.12 g/cm$^3$ and relative viscosity of 4.00 was supplied by Li Peng Enterprise (China).

The commercial virgin and post-processing vacuum bags for testing were made available by Alltec Materiais Compostos Ltd. This material is usually used for the manufacture of several composite parts for the aerospace industry. The post-processing vacuum bags used to prepare the blends with LLDPE came from heating cycle inside an autoclave, usually at 120 °C for 6 h. This heating cycle was necessary to cure the thermosetting resin of the composite.

The linear low-density polyethylene (LLDPE) of specification IC 32 was supplied by Braskem (Brazil) with a density of 0.924 g/cm$^3$ and melt flow index (MFI) of 29 g/10 min (190 °C/2.16 kg).

The maleic anhydride-grafted linear low-density polyethylene, LLDPE-*g*-MA, (Polybond®® 3109) with a density of 0.92 g/cm$^3$ and 1 wt.% of maleic anhydride was purchased from Crompton Corporation (USA). The LLDPE and LLDPE-*g*-MA present almost the same melt index (MFI = 30 g/10 min (190 °C/2.16 kg)).

### 2.2. Cryogenic Milling and Characterization of Post-Processing PA6

The post-processing PA6 films were cryogenically milled using liquid nitrogen, in a mini mill IKA, model A11. The milling occurred for 1 min. This procedure was necessary to obtain the material in powder, which is the strategy required for processing in the extruder.

Virgin and post-processing PA6 films from the vacuum bags were characterized by Fourier transform infrared spectroscopy (FTIR) using a Shimadzu spectrophotometer (model IRAffinity-1). Each spectrum obtained between 4000 and 400 cm$^{-1}$, corresponding to the average of 10 scans at a resolution of 4 cm$^{-1}$.

The viscosity at low shear rates was evaluated using a controlled stress TA instruments rheometer, model AR G2 with parallel plate geometry, 25 mm plate diameter and gap of 1 mm. The tests were performed in steady state, with an inert atmosphere of nitrogen and temperature of 220 °C.

The thermal characterization was evaluated by differential scanning calorimetry (DSC) using a Netzsch equipment, model 204 F1 Phoenix®®, under N$_2$ atmosphere. The samples (10 mg) were heated from room temperature to 250 °C at 10 °C min$^{-1}$ and kept at this temperature for 3 min to erase any previous thermal history, then cooled to 0 °C at 10 °C min$^{-1}$ to determine the crystallization temperature (T$_c$). After this, the samples were heated to 250 °C at 10 °C min$^{-1}$. The melting point (T$_m$) was evaluated from the endothermic peak of the second heating cycle. The degree of crystallinity (X$_c$) was calculated according to Equation (1):

$$X_c(\%) = \frac{\Delta H_m - \Delta H_c}{\Delta H_m{}^o} \times 100\% \tag{1}$$

where, $X_c(\%)$ is the degree of crystallinity, $\Delta H_m$ is the melting enthalpy obtained by DSC, $\Delta H_c$ is the crystallization enthalpy during heating that in this case is 0 J/g and $\Delta H_m{}^o$ is the theoretical melting heat value for 100% crystalline material (190.8 J/g for PA6 [21]).

### 2.3. Processing of PA6/LLDPE/LLDPE-g-MA Blends

All materials were dried in a vacuum oven at 80 °C for 24 h prior to processing. PA6/LLDPE/LLDPE-*g*-MA (90/5/5) blend was prepared using a co-rotational twin-screw extruder, AX Plasticos (model AX16:40DR) (L/D = 40, D = 16 mm). The temperature profile was 200, 220, 220,

220, 230 °C and the screw speed was set at 160 rpm. Using the same processing conditions, the neat PA6 was replaced by post-processing PA6 in the contents of 5, 10, 15, and 20 wt.%. Table 1 shows the compositions of the samples prepared in this work and the corresponding nomenclature. All the extrudates blends were pelletized at the die exit, and dried in a vacuum oven at 80 °C for 24 h. Standard test specimens were prepared using a hydro-pneumatic press from MH Equipamentos Ltd. (model PR8H), at 220 °C, with a pressing time of 5 min and a pressure of 5 bar.

**Table 1.** Compositions of the PA6/LLDPE/LLDPE-*g*-MA blends studied.

| | Post-Processing PA6 (wt.%) | PA6 (wt.%) | LLDPE (wt.%) | LLDPE-*g*-MA (wt.%) |
|---|---|---|---|---|
| PA6 | 0 | 100 | 0 | 0 |
| LLDPE | 0 | 0 | 100 | 0 |
| Blend | 0 | 90 | 5 | 5 |
| Blend (5% post-processing PA6) | 5 | 85 | 5 | 5 |
| Blend (10% post-processing PA6) | 10 | 80 | 5 | 5 |
| Blend (15% post-processing PA6) | 15 | 75 | 5 | 5 |
| Blend (20% post-processing PA6) | 20 | 70 | 5 | 5 |

### 2.4. Characterization of PA6/LLDPE/LLDPE-g-MA Blends

All blends were characterized by FT-IR, DSC and thermogravimetric analysis (TGA). The procedure and conditions were the same used in the analysis of PA6 films. The degree of crystallinity for each phase was calculated using an adaptation of Equation (1): $X_c$ (%) $- (\Delta H_m/(\Delta H_m^0 \times \phi_{blend})) \times 100$, where $\phi_{blend}$ is the mass fraction of the component in the blend and using the enthalpy of fusion of 100% crystalline polymer ($\Delta H_m^0$) of 190.8 J/g for PA6 and 140.6 J/g for LLDPE and LLDPE-*g*-MA [21]. Thermogravimetric analysis (TGA) of the blends was performed using a Netzsch model TG 209 F1 Iris$^{®®}$ equipment, from room temperature to 800 °C at a heating rate of 20 °C min$^{-1}$, under $N_2$ atmosphere.

Mechanical properties of the blends were evaluated by an Izod impact test performed on a CEAST/Instron Izod impact testing machine (model 9050), with a 2.75 J hammer. The test method adopted was carried out according to ASTM D256-06. All the test specimens were notched using a manual notched machine (CEAST/Instron), with a depth of 2.54 ± 0.1 mm.

The fracture surface morphology of the blends was analyzed by scanning electron microscopy (SEM). The samples were cryogenically fractured, placed on aluminum stubs, and covered with a thin layer of gold. The fracture surface was observed in a scanning electron microscope FEI Inspect S50, operating at 20 keV.

## 3. Results and Discussion

### 3.1. Characterization of Post-Processing PA6

Figure 1 shows the FT-IR spectra of the virgin and post-processing PA6 films from the vacuum bags, designated as PA6 films.

Analyzing the spectra of the virgin and post-processing PA6 films from vacuum bags, it was verified that there are no significant differences between them, where both have the same bands, which are characteristic of PA6. The main characteristic bands of PA6 are indicated on the FT-IR spectra. The band at 3300 cm$^{-1}$ is associated with the stretching of the NH group, at 1650 cm$^{-1}$, referring to the stretching of the CO group. The characteristic band at 1550 cm$^{-1}$ is associated with the stretching of the group NH and CN deformation and the band at 1300 cm$^{-1}$ is due to $(CH_2)_5$ groups. The band at 1135 cm$^{-1}$ is attributed to the complex vibration of the amide group and methylene chain and/or combination of NH deformation and O=CN stretching, and the band at 934 cm$^{-1}$ corresponding to the deformation of NH [22–24]. The similarity of the spectra indicates that the vacuum compression molding process did not cause significant chemical changes in the structure of the PA6 films. Su et al. [25] studied the influence of reprocessing on the mechanical properties and structure of PA6. The results showed that there were no

chemical changes in the FT-IR spectra of PA6 structure during the recycling process. Goitisolo et al. [26] studied the reprocessing of PA6 nanocomposites using injection molding for five times. In addition, no chemical changes in the FT-IR spectra of PA6 were observed.

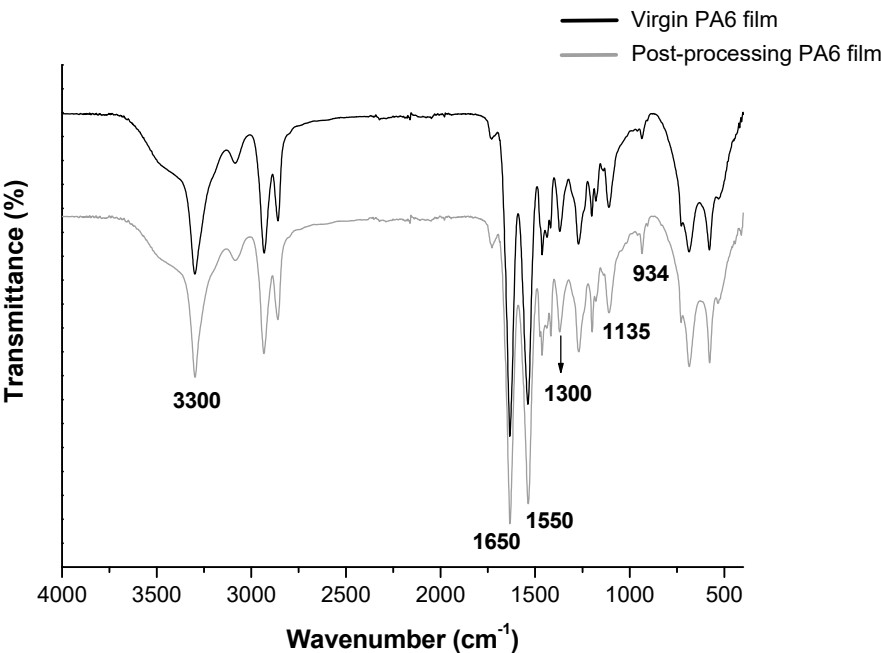

**Figure 1.** FT-IR spectra of the virgin and post-processing vacuum bags.

Figure 2 shows the DSC thermograms of the virgin and post-processing PA6 films from vacuum bags. Table 2 summarizes the values of glass transition temperature ($T_g$), melting temperature ($T_m$), and melting enthalpy ($\Delta H_m$) obtained during the first and second heating scans and the values of crystallization temperature ($T_c$) obtained during the cooling scan, in addition to the values of crystallinity degree ($X_c$).

In relation to the first heating, the post-processing PA6 film presented slightly higher $T_g$ and $T_m$ values and a lower degree of crystallinity compared to the virgin PA6 film. However analyzing the curves of the cooling and the second heating, there was practically no change in the values of $T_c$, $T_g$, $T_m$, and $X_c$, which indicates, therefore, that the vacuum compression molding process, to which the PA6 film was subjected, did not cause significant changes in the thermal properties of this material. In general, the recycling of the polyamide may cause chain breaks, resulting in a reduction of the molecular weight and an increase of the molecular weight distribution, which can consequently alter the thermal properties of the material [25,26]. Another consequence of the chain breaks is the creation of sites that favor the formation of crosslinks, which can reduce the crystallinity degree [27]. Therefore, as no changes were observed in these properties, it was concluded that the process did not cause any significant thermal degradation to the point of affecting the thermal properties of the PA6 film after the compression molding process.

Figure 3 shows the curves of viscosity as a function of the shear rate obtained by deformational rheometry of the virgin and post-processing PA6 films.

Analyzing the curves of virgin and post-processing PA6 films, it was verified that both films exhibit pseudoplastic behavior, where the viscosity decreases with increasing shear rate. The post-processing PA6 film presented higher initial viscosity ($\eta_0$) values ($\eta_0$ around 45,000 Pa.s) when compared to the virgin PA6 film with $\eta_0$ around 15,000 Pa.s. The $\eta_0$ was measured at the shear rate of 0.01 s$^{-1}$. The higher viscosity presented by the post-processing PA6 film can be explained possibly by the formation of crosslinks during the vacuum compression molding process. Since small crosslinking occurs between the PA6 chains, a consequent increase in viscosity occurs for the entire range of shear rate studied. The higher viscosity may also indicate that there was no significant thermal degradation during the

process, since chain breaks would result in a decrease in viscosity, which was not the case [25,26]. Analyzing the curves at low shear rates (0.01 to 0.1 s$^{-1}$) it can be observed that the post-processing PA6 film presents a narrower Newtonian plateau, therefore, even with higher viscosity values, its use is feasible, since the pseudoplastic behavior starts at lower shear rates, that is, the material will begin to flow at lower shear rates, facilitating the processing.

From these observations, it can be concluded that the post-processing vacuum bags have properties similar to the neat PA6, with thermal, chemical, and rheological characteristics that allow its use in substitution of neat material.

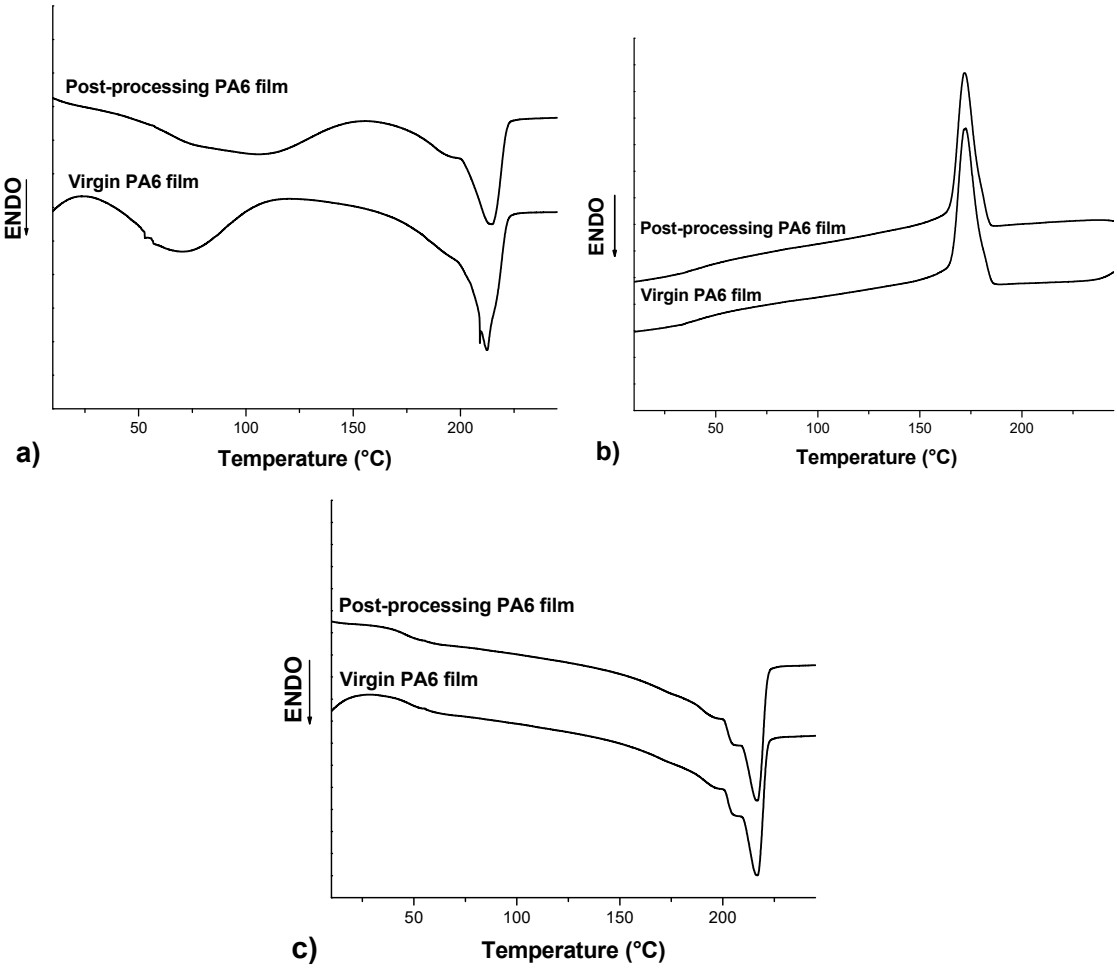

**Figure 2.** Differential scanning calorimetry (DSC) thermograms of the virgin and post-processing PA6 films. (**a**) First heating, (**b**) cooling, and (**c**) second heating.

**Table 2.** Values of $T_g$, $T_m$, $\Delta H_m$, and $X_c$ obtained during first and second heating scans and $T_c$ obtained during the cooling scan for the virgin and post-processing PA6 films.

| Samples | First Heating | | | | Cooling | | Second Heating | | |
|---|---|---|---|---|---|---|---|---|---|
| | $T_{g1}$ (°C) | $T_{m1}$ (°C) | $\Delta H_{m1}$ (J/g) | $X_{c1}$ (%) | $T_c$ (°C) | $T_{g2}$ (°C) | $T_{m2}$ (°C) | $\Delta H_{m2}$ (J/g) | $X_{c2}$ (%) |
| Virgin PA6 film | 53 | 212 | 48.07 | 25.2 | 172 | 48 | 217 | 42.00 | 22.0 |
| Post-processing PA6 film | 58 | 215 | 36.4 | 19.1 | 172 | 45 | 217 | 41.19 | 21.6 |

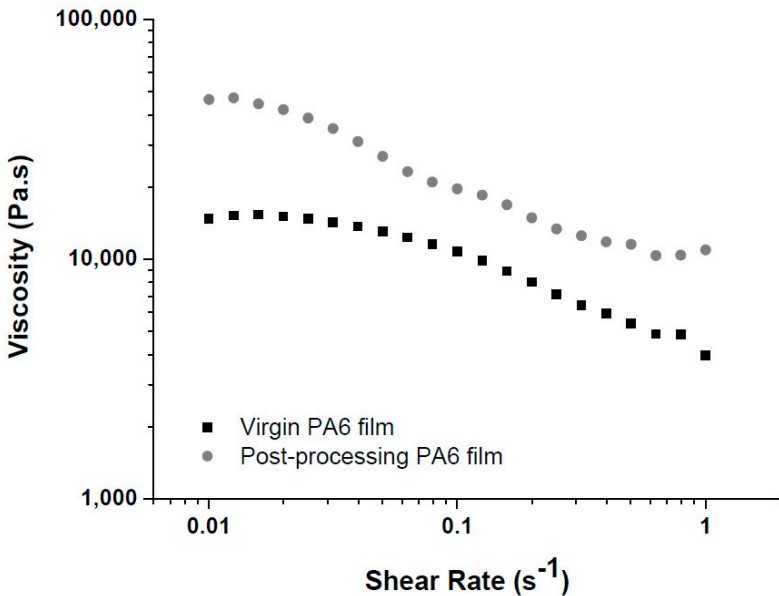

**Figure 3.** Curves of viscosity as a function of the shear rate obtained by deformational rheometry of the virgin and post-processing PA6 films.

*3.2. FT-IR, Thermal, Mechanical, and Morphological Characterization of PA6/LLDPE/LLDPE-g-MA Blends*

Figure 4 shows the FT-IR spectra of neat PA6, neat LLDPE and the PA6/LLDPE/LLDPE-*g*-MA blends with different contents of post-processing PA6.

Analyzing the spectra for all the blends it is verified that there are no differences between them, where the identified bands refer to the PA6, LLDPE, and the compatibilizer agent (LLDPE-*g*-MA). The PA6 characteristic bands were described previously (3300 cm$^{-1}$, 1650 cm$^{-1}$, 1550 cm$^{-1}$, 1300 cm$^{-1}$, 1135 cm$^{-1}$ e 934 cm$^{-1}$). In relation to the LLDPE phase, the characteristic bands are around 2970 and 2870 cm$^{-1}$, due to the asymmetric and symmetrical axial deformation of the -CH$_2$, the band at 1473 cm$^{-1}$ referring to the angular deformation -CH, the band at 1300 cm$^{-1}$ corresponding to the asymmetric angular deformation of the methyl group, and the band at 731 cm$^{-1}$ corresponding to the symmetrical angular deformation of the -CH$_2$ [28]. Regarding the compatibilizer agent, a band around 1220 cm$^{-1}$ was observed, which refers to the axial deformation of the group -O-C=O given by the presence of the maleic anhydride. However, no bands were observed for the -C-N- bonds of the imide groups (1374 cm$^{-1}$), which can be formed by the reaction between the terminal group of PA6 and the compatibilizer agent LLDPE-*g*-MA, which would be responsible for the formation of the LLDPE-*g*-MA-*g*-PA6 copolymer at the interface, increasing the interfacial adhesion of the blend [6]. In this way, it can be concluded that there was no effective compatibilization of the blend by this reaction, only by steric hindrance mechanisms, which cause elastic repulsion of the chains and prevent the coalescence of the LLDPE phase given by the presence of maleic anhydride.

Figure 5 shows the DSC thermograms of neat PA6, neat LLDPE, and the blends. Table 3 shows the values of the thermal properties during the first heating and Table 4 shows the values for the cooling, second heating and $T_{onset}$ obtained by TGA.

Analyzing the curves of heating for all the blends it can be observed two melting temperatures ($T_m$) referring to the PA6 phase and LLDPE phase. These observations may be indicative of the immiscibility of these constituents in the blend. The $T_m$ values obtained were similar to the values of the neat polymers (220 °C for PA6 and 120 °C for LLDPE [5]) for all the compositions. However, different values of $T_g$ corresponding to PA6 were observed in the first heating, with values varying from 28 °C (blends with 15% post-processing PA6) to 42 °C (blends with 10% post-processing PA6). These differences can be explained by the thermal history given by the process of material extrusion and the absorption of water, where the water molecules can act as plasticizer, decreasing the $T_g$ value. These

results are confirmed by analyzing the second heating where the $T_g$ values of PA6 for all compositions are close to each other, with values around 55 °C. In relation to cooling, an increase of $T_c$ of PA6 was observed with the addition of post-processing PA6, from 187 °C to 191 °C, with values decreasing according to the higher content of post-processing PA6 added. The $T_c$ of the LLDPE phase decreased with the addition of post-processing PA6. In relation to the degree of crystallinity, it was observed in the first heating, that the degree of crystallinity of the PA6 phase in the blend showed no significant changes, presenting values around 30.9%. In relation to the degree of crystallinity of the LLDPE phase, it was observed that the addition of post-processing PA6 favored the crystallization of this phase, with values increasing progressively from 31.5% (blends without post-processing PA6) up to 52.5% (blends with 20% post-processing PA6), indicating, therefore, that post-processing PA6 can act as a nucleating agent for the LLDPE phase. This increase in the degree of crystallinity of the LLDPE phase with the addition of post-processing PA6 makes this material stiffer reducing its toughening effect in the blend.

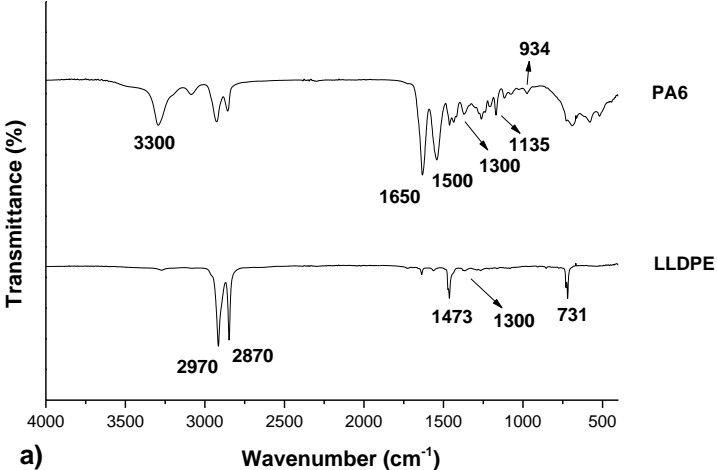

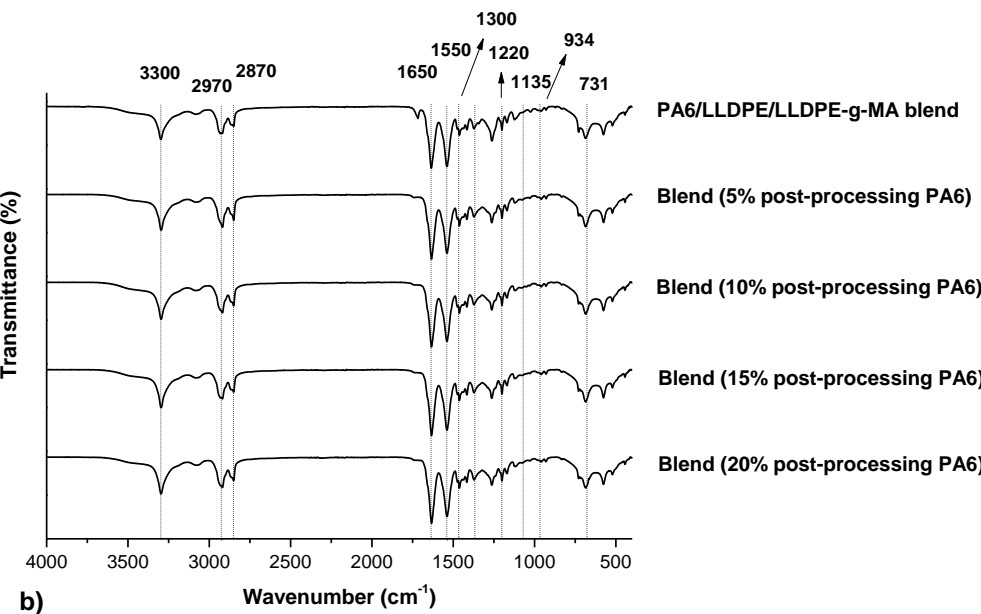

**Figure 4.** FT-IR spectra of (**a**) neat PA6, neat LLDPE and (**b**) the PA6/LLDPE/LLDPE-*g*-MA blends with different post-processing PA6 contents.

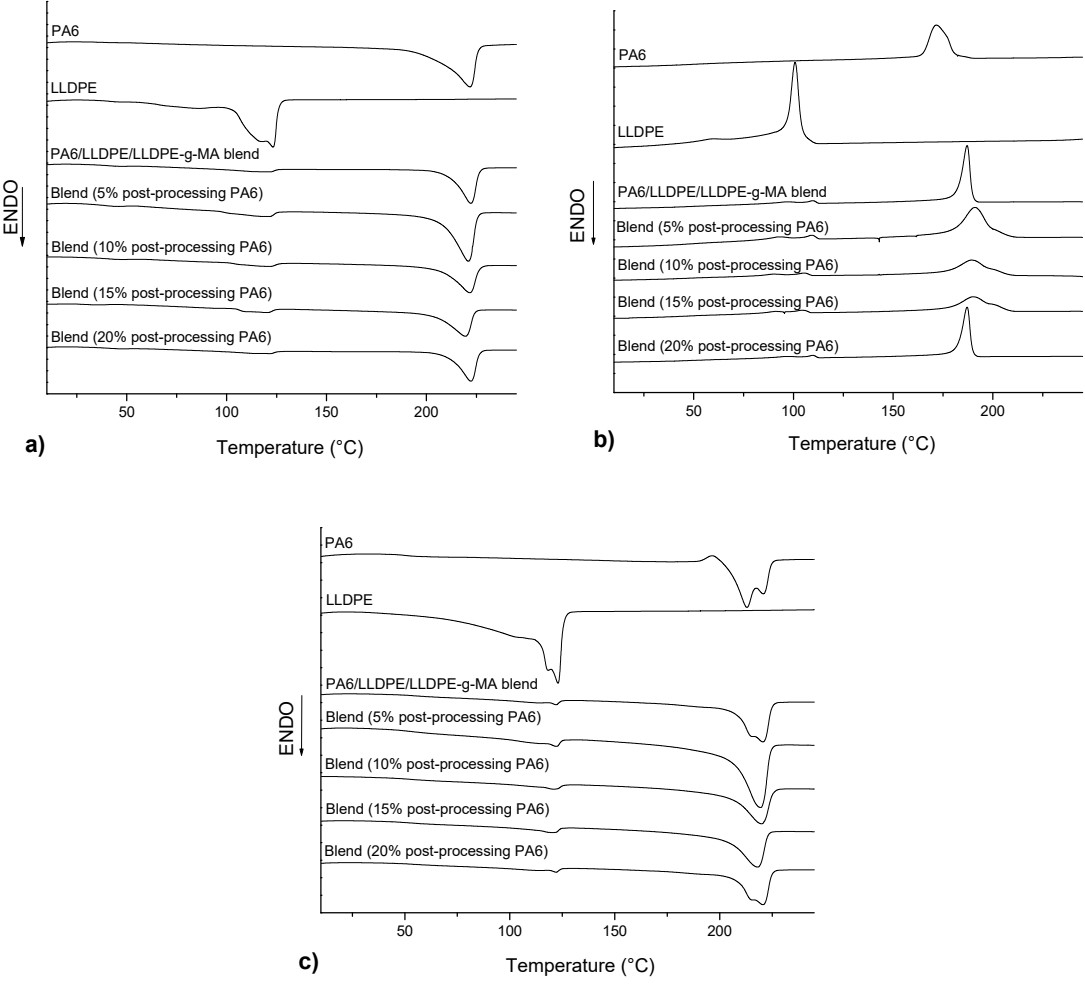

**Figure 5.** DSC thermograms of neat PA6, neat LLDPE and the blends. (**a**) First heating, (**b**) cooling and (**c**) second heating.

**Table 3.** Values of $T_g$, $T_m$, $\Delta H_m$ and $X_c$ of neat PA6, neat LLDPE and the blends obtained during first heating.

| Samples | First Heating | | | | | | |
|---|---|---|---|---|---|---|---|
| | $T_{g1\ PA6}$ (°C) | $T_{m1\ PA6}$ (°C) | $\Delta H_{m1\ PA6}$ (J/g) | $Xc_{1\ PA6}$ (%) | $T_{m1\ LLDPE}$ (°C) | $\Delta H_{m1\ LLDPE}$ (J/g) | $X_{c1\ LLDPE}$ (%) |
| PA6 | 62.8 | 222 | 82.05 | 43.0 | — | — | — |
| LLDPE | — | — | — | — | 123 | 87.91 | 62.5 |
| Blend | 40 | 222 | 48.37 | 28.2 | 121 | 4.43 | 31.5 |
| Blend (5% post-processing PA6) | 37 | 221 | 55.67 | 32.4 | 121 | 6.14 | 43.7 |
| Blend (10% post-processing PA6) | 42 | 222 | 53.15 | 30.9 | 121 | 7.09 | 50.4 |
| Blend (15% post-processing PA6) | 28 | 220 | 57.26 | 33.3 | 120 | 7.20 | 51.2 |
| Blend (20% post-processingPA6) | 40 | 221 | 51.67 | 30.1 | 121 | 7.38 | 52.5 |

**Table 4.** Values of $T_c$ obtained during cooling, $T_g$, $T_m$, $\Delta H_m$, and $X_c$ obtained during second heating and $T_{onset}$ obtained by TGA.

| Samples | Cooling | | | | Second Heating | | | | | $T_{onset}$ (°C) |
|---|---|---|---|---|---|---|---|---|---|---|
| | $T_{c\ PA6}$ (°C) | $T_{c\ LLDPE}$ (°C) | $T_{g2\ PA6}$ (°C) | $T_{m2\ PA6}$ (°C) | $\Delta H_{m2\ PA6}$ (J/g) | $X_{c2\ PA6}$ (%) | $T_{m2\ LLDPE}$ (°C) | $\Delta H_{m2\ LLDPE}$ (J/g) | $X_{c2\ LLDPE}$ (%) | |
| PA6 | 172 | — | 51.2 | 213 | 50.94 | 26.7 | — | — | — | 435.4 |
| LLDPE | — | 101 | — | — | — | — | 123 | 63.47 | 45.1 | 453.5 |
| Blend | 187 | 110 | 55 | 221 | 54.24 | 31.6 | 122 | 0.83 | 5.9 | 430.7 |
| Blend (5% post-processing PA6) | 191 | 110 | 53 | 220 | 64.12 | 37.3 | 123 | 0.99 | 7.1 | 431.2 |
| Blend (10% post-processing PA6) | 190 | 106 | 52 | 220 | 55.39 | 32.2 | 121 | 1.98 | 14.1 | 425.0 |
| Blend (15% post-processing PA6) | 190 | 105 | 56 | 218 | 55.15 | 32.1 | 121 | 2.93 | 20.8 | 421.4 |
| Blend (20% post-processing PA6) | 189 | 104 | 56 | 219 | 52.16 | 30.4 | 122 | 4.06 | 28.9 | 426.7 |

Figure 6 shows the TGA curves of neat PA6, neat LLDPE and PA6/LLDPE/LLDPE-*g*-MA blends with different contents of post-processing PA6.

The neat LLDPE has a greater $T_{onset}$ than neat PA6, which in turn presents a mass loss at 150 °C corresponding to humidity loss. Analyzing the curves and $T_{onset}$ for the blends, no significant differences were observed between the values, with an average value of 427 °C, indicating, therefore, that the addition of post-processing PA6 did not significantly influence the thermal degradation behavior of the blends. Cataño et al. [10] studied the degradation behavior of the PA6/LLDPE blends compatibilized with SEBS-*g*-DEM. In the results was also observed a single step thermal decomposition curve, and there was no significant difference between the blends.

Figure 7 shows the Izod impact strength of the neat PA6 and the PA6/LLDPE/LLDPE-*g*-MA blends with different contents of post-processing PA6. It is noteworthy that the neat LLDPE did not break during the test, even using different hammers.

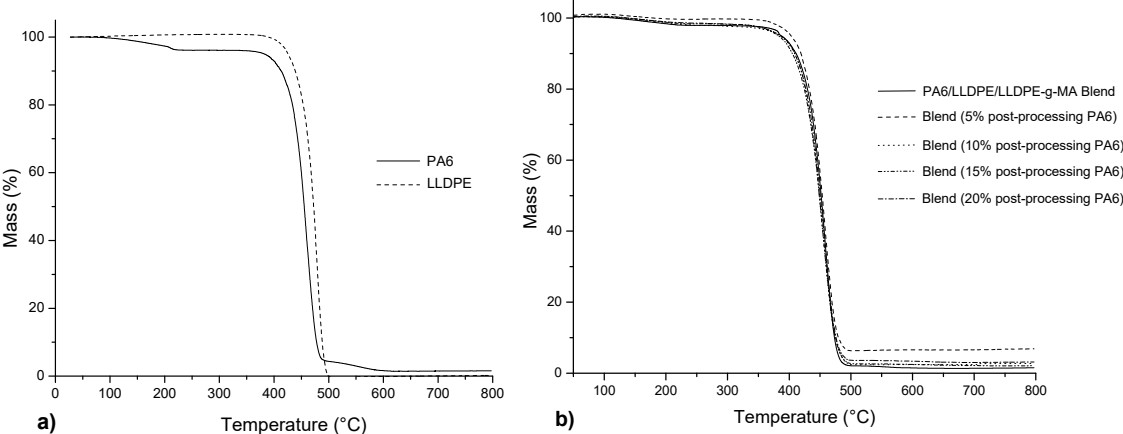

**Figure 6.** Thermogravimetric analysis (TGA) curves of (**a**) neat PA6 and neat LLDPE and (**b**) PA6/LLDPE/LLDPE-*g*-MA blends with different contents of post-processing PA6.

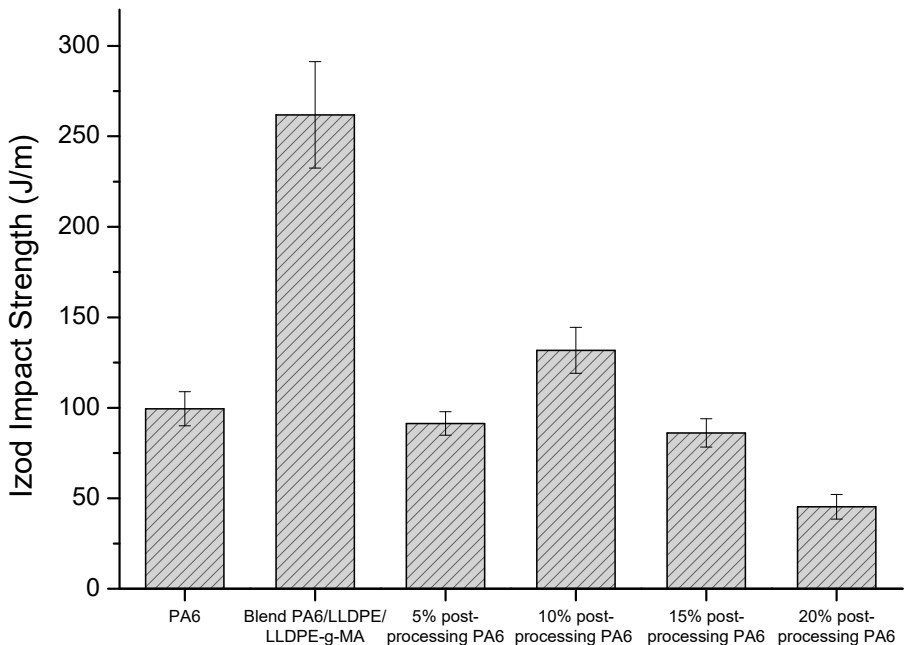

**Figure 7.** Izod impact strength of the neat PA6 and the PA6/LLDPE/LLDPE-*g*-MA blends with different contents of post-processing PA6.

Analyzing the results, the PA6/LLDPE/LLDPE-*g*-MA blend shows higher impact strength when compared to the neat PA6, which was expected due to the toughening process given by the incorporation of the LLDPE as a second phase into the PA6 phase. However, it is observed that the addition of the post-processing PA6 in the composition of the blends significantly reduced the impact strength, where only for the composition with the addition of 10 wt.% of post-processing PA6, which presented impact strength slightly above neat PA6. Therefore, the addition of post-processing PA6 increased the brittleness of the blends. This fact can be explained by analyzing the degree of crystallinity of the phases, where it is verified that the addition of post-processing PA6 favored the crystallization of the LLDPE phase in the blend, damaging the toughening process. In this way, an increase in the crystallinity degree of the LLDPE phase cause the increase in the stiffness of this phase in the blend. Thus, instead of the LLDPE phase acting as a toughening agent, it acts as a reinforcing agent.

The performance of post-processing PA6 as nucleating agent of the LLDPE phase in the PA6/LLDPE blends is interesting from the technological point of view, and may be an alternative to increase the stiffness of this phase when required in specific applications.

Figure 8 shows SEM micrographs of PA6/LLDPE/LLDPE-*g*-MA blends with different post-processing PA6 contents.

In Figure 8a it is possible to observe the PA6 matrix and the LLDPE second phase in the form of spherical particles, indicated by the white arrows. For the blend without addition of post-processing PA6, a homogeneous morphology is observed with second phase particles of homogeneous size and well dispersed and distributed into the matrix. Some cavities from the extraction of the second phase particles are also observed. Analyzing the blends with 5 wt.% of post-processing PA6 (Figure 8b) it is possible to observe the well-distributed second phase particles into the matrix with homogeneous size. For the blends with 10, 15, and 20 wt.% of post-processing PA6, the second phase particles were not clearly visible, only the visualization of some of them being possible. Kelar e Jurkowski [20] and Das et al. [6] obtained similar results to this work. The LLDPE-*g*-MA promoted a significant improvement in the interfacial adhesion between the blend components, decreasing the second phase particle size and improving its dispersion. With the stable morphology obtained, there were improvements in the physico-mechanical properties of the compatibilized blends.

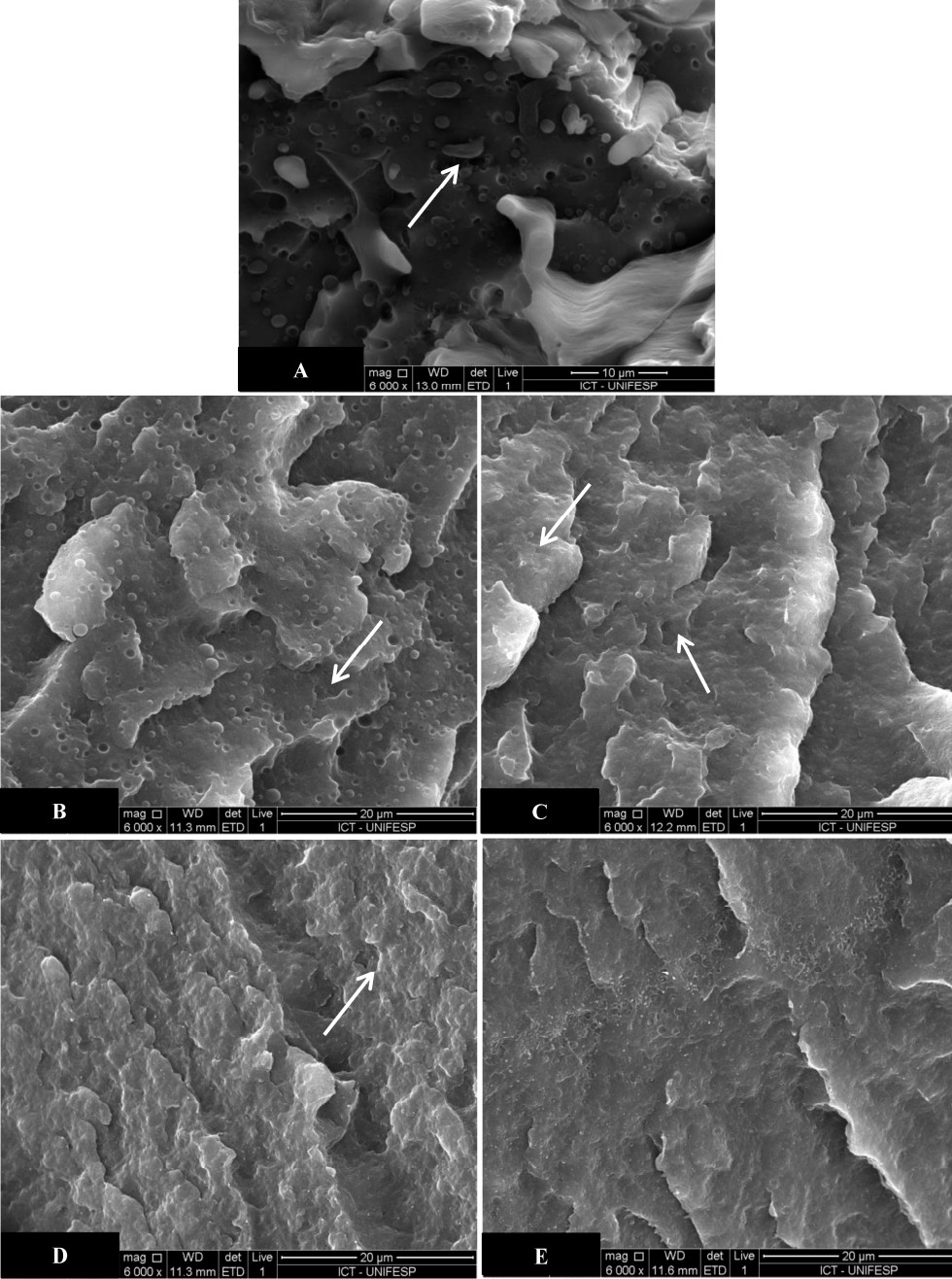

**Figure 8.** SEM micrographs of PA6/LLDPE/LLDPE-*g*-MA blends with different post-processing PA6 contents (**a**) 0 wt.%, (**b**) 5 wt.%, (**c**) 10 wt.%, (**d**) 15 wt.% e (**e**) 20 wt.%.

In order to exemplify the effect of the compatibilizer agent, the SEM micrographs of the PA6/LLDPE blend (90/10), without addition of compatibilizer agent and PA6/LLDPE/LLDPE-*g*-MA blends (90/5/5) are shown in the Figure 9. In these SEM micrographs it is possible to observe that the addition of the compatibilizer agent decreased the particle size of the LLDPE phase and makes the morphology more homogeneous.

Therefore, the morphology of the blends shows the occurrence of the compatibilization given by the presence of the LLDPE-*g*-MA and the Izod impact results show that the lower impact strength of the blends is explained exclusively by the greater crystallinity of the LLDPE phase caused by the presence of the post-processing PA6.

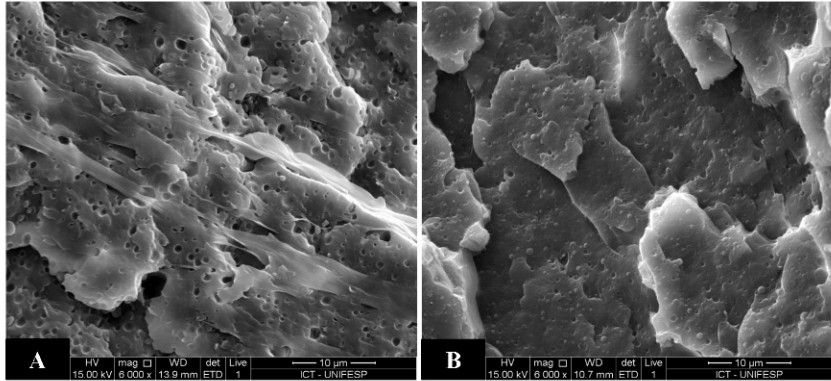

**Figure 9.** SEM micrographs of (**a**) PA6/LLDPE (90/10) and (**b**) PA6/LLDPE/LLDPE-*g*-MA (90/5/5).

## 4. Conclusions

The use of PA6 post-processing vacuum bags proved to be a very innovative alternative for the production of PA6/LLDPE blends contributing to the reduction of waste from the aerospace industry. The PA6 post-processing vacuum bags had the same thermal, chemical, and rheological properties as virgin vacuum bags and were therefore eligible for replacement of neat PA6 in polymer blends. The new strategy presented consists of the cryogenic milling of these materials after consumption and addition directly in the extrusion process to prepare the polymer blends. Regarding the chemical characterization of the blends with addition of post-processing PA6, no changes were observed when compared to the polymer blend PA6/LLDPE/LLDPE-*g*-MA. The compatibilization effect of the blends was verified by the morphological analysis, being possible to observe the PA6 matrix and the LLDPE second phase in the form of spherical particles of homogeneous size and well dispersed into the matrix. Thermal characterization showed that the addition of post-processing PA6 favored the crystallization of the LLDPE phase in the blends. Due to the higher stiffness, the LLDPE phase stopped acting as a toughening agent and started acting as a nucleating agent to the LLDPE phase for the PA6/LLDPE blend. These results are confirmed by the analysis of the impact strength of the blends with addition of post-processing PA6, where there was a significant decrease of the values in comparison to the blend without addition of post-processing PA6. Reducing the waste generated for manufacturing aerospace products coupled with the use of this waste material as a nucleating agent for the LLDPE phase in PA6/LLDPE blends can be a profitable and sustainable alternative for this industry.

**Author Contributions:** Conceptualization, G.P.M.d.S. and F.R.P.; methodology, G.P.M.d.S.; E.G.R.d.A.; L.S.M.; O.F. and F.R.P.; validation, L.S.M.; O.F. and F.R.P.; formal analysis, G.P.M.d.S.; E.G.R.d.A. and L.S.M.; investigation, G.P.M.d.S.; E.G.R.d.A.; L.S.M.; O.F. and F.R.P.; resources, X.X.; data curation, G.P.M.d.S.; E.G.R.d.A.; L.S.M.; O.F. and F.R.P.; writing—original draft preparation, G.P.M.d.S.; E.G.R.d.A.; L.S.M.; O.F. and F.R.P.; writing—review and editing, F.R.P.; visualization, G.P.M.d.S.; E.G.R.d.A.; L.S.M.; O.F. and F.R.P.; supervision, F.R.P.; project administration, F.R.P.; funding acquisition, F.R.P.

**Funding:** This research received no external funding.

**Acknowledgments:** The authors would like to thank the Brazilians Funding Institutions CNPq (Conselho Nacional de Desenvolvimento Científico e Tecnológico—Process 405675/2018-6 and 310196/2018-3) and FAPESP (Fundação de Amparo à Pesquisa do Estado de São Paulo—Process 2016/19978-9) for financial support. The authors also thank Master Polymers for the donating of PA6.

**Conflicts of Interest:** The authors declare that they have no conflict of interest.

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
