# Peer review of "A New Strategy for the Use of Post-Processing Vacuum Bags from Aerospace Supplies: Nucleating Agent to LLDPE Phase in PA6/LLDPE Blends"

_recycling, doi:10.3390/recycling4020018_

Round 1
Reviewer 1 Report
REVIEW
Title: A new strategy for the use of post-processing vacuum bag from aerospace supplies: nucleating agent to LLDPE phase in PA6/LLDPE blends
The manuscript described a way to use the PA6 wastes from aerospace industry in obtaining new materials that can be reused or used elsewhere.
The manuscript can be published after minor revision, as I suggest in the rows below:
p.1, r.38: “…a stack of reinforcement with thermosetting resin…” is not clear.
p.2, r.80: “…a decrease in the particle size of the second phase…”
p.2, r. 92: “Another goal is to study the effect of…”
p.3, r. 103: “…usually at 1200C…”
p.3, r.109: I would say that the polymers present “almost the same” MFI, as 29g/10 min is different from 30g/10 min.
p.3, r.128: Tm “was taken at the melting endothermic peak”. Maybe “was evaluated from the endothermic peak” is better.
p.4, r.142: “Table 1 shows the composition of samples prepared in this work and the corresponding nomenclature”
p.4, r.154: “The degree of crystallinity for each phase was calculated…”
p.4, r.155: The adaptation of equation (1)should be Xc(%) = (DHm/(DHm0 x fblend)) x 100
p.5 For the assignment of FTIR bands the authors used only one reference, 22. In other papers the band from 1300 cm-1 represents the amide III band, while bending of CH2 is around 1400 cm-1. I suggest to the authors to check more references for FTIR spectra, if they think these bands are really important.
p.6, r.201: “However analyzing the curves of the cooling and the second heating, practically no changes in the values of Tc, Tg, Tm, and Xc can be observed, which….”
p.7 The rheological behavior of neat and post-processing PA6 is not clearly explained. First, were the Newtonian viscosities (h0) evaluated using a model or they were the values at 0.01 s-1? The authors explain the higher viscosity of the post-processing polymer by “small contamination with epoxy resin”. But such contamination wouldn’t be noticed in FTIR spectra? Maybe the authors would like to consider the formation of crosslinks during processing, which will explain the higher viscosity on the entire range of shear rates and the lower degree of crystallinity for the processed polymer. Also, the post-processing PA6 film does not present “a lower Newtonian plateau” but a shorter or narrower one. Moreover, “the material will begin to flow at lower shear rates, facilitating the processing.”
p.10, r.280: “…increasing progressively from 31.5%...”
p.11, r.290: “The neat LLDPE has a greater Tonset than neat PA6, which in turn presents a mass loss at 1500C corresponding to humidity loss.”
p.12, r.304: “…due to the toughening process given by the incorporation of the LLDPE as a second phase.”
p.13, r.353: “…well dispersed into the matrix.”
p.13, r.354: Please reformulate the entire phrase because “favored the crystallization” and “increase in the crystallinity degree” seams the same think.
p.14, r.360: “The reduction of aerospace waste added to the use of this material as a nucleating agent…” must be reformulated more clearly.
Author Response
Response to Reviewer 1
R: We are so grateful for valuable comments about our study. All you contributions are been accepted.
The manuscript described a way to use the PA6 wastes from aerospace industry in obtaining new materials that can be reused or used elsewhere.
The manuscript can be published after minor revision, as I suggest in the rows below:
p.1, r.38: “…a stack of reinforcement with thermosetting resin…” is not clear.
R: The sentence was modified to make it clearer.
p.2, r.80: “…a decrease in the particle size of the second phase…”
R: The sentence was corrected
p.2, r. 92: “Another goal is to study the effect of…”
R: The sentence was corrected
p.3, r. 103: “…usually at 1200C…”
R: The sentence was corrected
p.3, r.109: I would say that the polymers present “almost the same” MFI, as 29g/10 min is different from 30g/10 min.
R: The sentence was corrected
p.3, r.128: Tm “was taken at the melting endothermic peak”. Maybe “was evaluated from the endothermic peak” is better.
R: The sentence was corrected
p.4, r.142: “Table 1 shows the composition of samples prepared in this work and the corresponding nomenclature”
R: The sentence was corrected
p.4, r.154: “The degree of crystallinity for each phase was calculated…”
R: The sentence was corrected
p.4, r.155: The adaptation of equation (1)should be Xc(%) = (DHm/(DHm0 x fblend)) x 100
R: The equation was corrected
p.5 For the assignment of FTIR bands the authors used only one reference, 22. In other papers the band from 1300 cm-1 represents the amide III band, while bending of CH2 is around 1400 cm-1. I suggest to the authors to check more references for FTIR spectra, if they think these bands are really important.
R: Two more new references were added that prove the same bands are present in PA6.
p.6, r.201: “However analyzing the curves of the cooling and the second heating, practically no changes in the values of Tc, Tg, Tm, and Xc can be observed, which….”
R: The sentence was corrected
p.7 The rheological behavior of neat and post-processing PA6 is not clearly explained. First, were the Newtonian viscosities (h0) evaluated using a model or they were the values at 0.01 s-1? The authors explain the higher viscosity of the post-processing polymer by “small contamination with epoxy resin”. But such contamination wouldn’t be noticed in FTIR spectra? Maybe the authors would like to consider the formation of crosslinks during processing, which will explain the higher viscosity on the entire range of shear rates and the lower degree of crystallinity for the processed polymer. Also, the post-processing PA6 film does not present “a lower Newtonian plateau” but a shorter or narrower one. Moreover, “the material will begin to flow at lower shear rates, facilitating the processing.”
R: We agreed with the reviewer. This part of the discussion has been reviewed and improved.
p.10, r.280: “…increasing progressively from 31.5%...”
R: The sentence was corrected
p.11, r.290: “The neat LLDPE has a greater Tonset than neat PA6, which in turn presents a mass loss at 1500C corresponding to humidity loss.”
R: The sentence was corrected
p.12, r.304: “…due to the toughening process given by the incorporation of the LLDPE as a second phase.”
R: The sentence was corrected
p.13, r.353: “…well dispersed into the matrix.”
R: The sentence was corrected
p.13, r.354: Please reformulate the entire phrase because “favored the crystallization” and “increase in the crystallinity degree” seams the same think.
R: The sentence was corrected
p.14, r.360: “The reduction of aerospace waste added to the use of this material as a nucleating agent…” must be reformulated more clearly.
R: The sentence was modified to make it clearer.
Reviewer 2 Report
In this study, the authors prepared and characterized the PA6/LLDPE/LLDPE-g-MA (90/5/5) blends with
the neat PA6 replaced by different percentage of post-processing PA6, which came from the plastic
vacuum bag, a common waste in the aerospace industry. The authors demonstrated that the properties
of post-processing PA6 were similar to the virgin PA6. Besides, the post-processing PA 6 helps to
increase the crystallinity degree of the LLDPE phase in the blends. The research provide a potential way
for recycle plastic vacuum bag made by PA6 in aerospace industry. However, there are some concerns
that need to be addressed before it can be published.
(1) There are some typos and format problems in the paper. For example, line 33, the “aggravated?”. In
line 115-116, line 148 and line 236-237, the fonts are not consistent. Others such as Tg and Tm should
be Tg and Tm (subscript).
(2) Some contents are repeated in the paper. For example, the same FTIR and DSC information has been
provided in line 148-157, and 115-118 and 123-135. Please consider to combine them. Besides, in line
327-335, same contents have been introduced in the introduction part (line 84-86, 79-81). There is no
need to repeat it again.
(3) Line 220-222, the authors mentioned “The higher viscosity presented by the post-processing PA6 film
can be explained by the small contamination with polymer resin (epoxy resin) from the vacuum
compression molding process.” Is there any direct evidence for the existence of such a contaminant?
Since there is no purification process of post-processing PA6 was included in the study, how the polymer
resin contaminant will affect the properties of the blends if it did exist?
(4) Line 250-252, it was concluded “there was no effective compatibilization of the blend by this reaction,
only by steric hindrance mechanisms given by the presence of maleic anhydride.” What the “steric
hindrance mechanisms” means here? How could the “steric hindrance mechanisms given by the
presence of maleic anhydride” can be seen by FTIR?
(5) Line 339-340, it was mentioned “the morphology of the blends shows the occurrence of the
compatibilization given by the presence of the LLDPE-g-MA”. However, the SEM images in Figure 8 were
about the blends with same 5% of LLDPE-g-MA but different post-processing PA6 contents. Since there
is no SEM image of the blend without LLDPE-g-MA or with different concentrations of LLDPE-g-MA, it is
hard to understand how the conclusion was drawn from.
(6) Line 356 to 358, “Due to the higher stiffness, the LLDPE phase stopped acting as a toughening agent
and started acting as a reinforcing agent and/or nucleating agent to the LLDPE phase” seems confused.
Do you mean “nucleating agent to the blend”?
Author Response
Response to Reviewer 2
R: We are so grateful for valuable comments about our study. All you contributions are been accepted.
In this study, the authors prepared and characterized the PA6/LLDPE/LLDPE-g-MA (90/5/5) blends with the neat PA6 replaced by different percentage of post-processing PA6, which came from the plastic vacuum bag, a common waste in the aerospace industry. The authors demonstrated that the properties of post-processing PA6 were similar to the virgin PA6. Besides, the post-processing PA 6 helps to increase the crystallinity degree of the LLDPE phase in the blends. The research provide a potential way for recycle plastic vacuum bag made by PA6 in aerospace industry. However, there are some concerns that need to be addressed before it can be published.
(1) There are some typos and format problems in the paper. For example, line 33, the “aggravated?”. In line 115-116, line 148 and line 236-237, the fonts are not consistent. Others such as Tg and Tm should be Tg and Tm (subscript).
R: All these points have been revised.
(2) Some contents are repeated in the paper. For example, the same FTIR and DSC information has been provided in line 148-157, and 115-118 and 123-135. Please consider to combine them. Besides, in line 327-335, same contents have been introduced in the introduction part (line 84-86, 79-81). There is no need to repeat it again.
R: The text has been revised and repeated parts have been removed or modified.
(3) Line 220-222, the authors mentioned “The higher viscosity presented by the post-processing PA6 film can be explained by the small contamination with polymer resin (epoxy resin) from the vacuum compression molding process.” Is there any direct evidence for the existence of such a contaminant? Since there is no purification process of post-processing PA6 was included in the study, how the polymer resin contaminant will affect the properties of the blends if it did exist?
R: The text has been modified. We have been working with this material for some time, we are carrying out complementary tests and so far there is no evidence of waste or epoxy resin. However, because of the time the vacuum bag is left in the high temperature process, it is possible that crosslinking occurs between the PA6 chains. This fact explains the increase in viscosity for the entire range of shear rate studied and also the decrease of the crystallinity of this material.
(4) Line 250-252, it was concluded “there was no effective compatibilization of the blend by this reaction,only by steric hindrance mechanisms given by the presence of maleic anhydride.” What the “steric hindrance mechanisms” means here? How could the “steric hindrance mechanisms given by the presence of maleic anhydride” can be seen by FTIR?
R: The mechanism was better explained in the text. It is not possible to detect by FTIR, but it is possible to observe in micrographs by decreasing the size of the LLDPE phase.
(5) Line 339-340, it was mentioned “the morphology of the blends shows the occurrence of the
compatibilization given by the presence of the LLDPE-g-MA”. However, the SEM images in Figure 8 were about the blends with same 5% of LLDPE-g-MA but different post-processing PA6 contents. Since there is no SEM image of the blend without LLDPE-g-MA or with different concentrations of LLDPE-g-MA, it is hard to understand how the conclusion was drawn from.
R: We agree with this placement. We added a new figure of this sample with and without compatibilizer agent to exemplify the effect.
(6) Line 356 to 358, “Due to the higher stiffness, the LLDPE phase stopped acting as a toughening agent and started acting as a reinforcing agent and/or nucleating agent to the LLDPE phase” seems confused.Do you mean “nucleating agent to the blend”?
R: The sentence was corrected. The correct sentence is: nucleating agent for the LLDPE phase in the PA6/LLDPE blend.
Reviewer 3 Report
The authors present a paper entitled “A new strategy for the use of post-processing vacuum 2 bag from aerospace supplies: nucleating agent to 3 LLDPE phase in PA6/LLDPE blends”, in which they introduce post-use PA6 in PA6/LLDPE blends and evaluate the obtained performance.
The paper presents interesting results and deserve to be published. My only concern is regarding the impact characterization. The authors claim that the performance peak at 10% post processing PA6 is due to the degree of crystallinity of the phases, which in principle can be correct, but what if the 5% post processing material has some experimental problem and the resulting impact strength is significantly lower than the supposed one? If the real performance of this sample would be higher (say 150-200 J/m), there would be a easy trend (the higher post-use material, the lower the properties) to discuss about. I would suggest to the authors to check it again.
Author Response
Response to Reviewer 3
The authors present a paper entitled “A new strategy for the use of post-processing vacuum 2 bag from aerospace supplies: nucleating agent to 3 LLDPE phase in PA6/LLDPE blends”, in which they introduce post-use PA6 in PA6/LLDPE blends and evaluate the obtained performance.
The paper presents interesting results and deserve to be published. My only concern is regarding the impact characterization. The authors claim that the performance peak at 10% post processing PA6 is due to the degree of crystallinity of the phases, which in principle can be correct, but what if the 5% post processing material has some experimental problem and the resulting impact strength is significantly lower than the supposed one? If the real performance of this sample would be higher (say 150-200 J/m), there would be a easy trend (the higher post-use material, the lower the properties) to discuss about. I would suggest to the authors to check it again.
R: We are so grateful for valuable comments about our study. All you contributions are been accepted.
We agree with your point of view. We did a check on both the results and the samples that were tested and improved the text.
Round 2
Reviewer 2 Report
Agree to publish.